# Preparation of 1-Hydroxy-2,5-hexanedione from HMF by the Combination of Commercial Pd/C and Acetic Acid

**DOI:** 10.3390/molecules25112475

**Published:** 2020-05-27

**Authors:** Yanliang Yang, Dexi Yang, Chi Zhang, Min Zheng, Ying Duan

**Affiliations:** 1Henan Key Laboratory of Function-Oriented Porous Material, College of Chemistry and Chemical Engineering, Luoyang Normal University, Luoyang 471934, China; dexiyang2000@163.com (D.Y.); zhangchi0826@126.com (C.Z.); zhengmin@lynu.edu.cn (M.Z.); 2College of Food and Drug, Luoyang Normal University, Luoyang 471934, China; duanying@mail.ustc.edu.cn

**Keywords:** HMF, 1-hydroxy-2,5-hexanedione, acetic acid, Pd/C, high concentration

## Abstract

The development of a simple and durable catalytic system for the production of chemicals from a high concentration of a substrate is important for biomass conversion. In this manuscript, 5-hydroxymethylfurfural (HMF) was converted to 1-hydroxy-2,5-hexanedione (HHD) using the combination of commercial Pd/C and acetic acid (AcOH) in water. The influence of temperature, H_2_ pressure, reaction time, catalyst amount and the concentration of AcOH and HMF on this transformation was investigated. A 68% yield of HHD was able to be obtained from HMF at a 13.6 wt% aqueous solution with a 98% conversion of HMF. The resinification of intermediates on the catalyst was characterized to be the main reason for the deactivation of Pd/C. The reusability of the used Pd/C was studied to find that most of the activity could be recovered by being washed in hot tetrahydrofuran.

## 1. Introduction

The production of chemicals from renewable resources is one of the targets for green and sustainable chemistry. Glucose, fructose and other carbohydrates, which are renewable every year in nature, are promising candidates for renewable resources [1,2,3]. The carbohydrate compounds can be readily dehydrated in the presence of acid to form 5-hydroxymethylfurfural (HMF), levulinic acid or lactic acid as critical platform compounds [4,5,6,7,8,9,10,11,12,13,14,15,16,17]. As a result, the conversion of HMF to valuable products has attracted much attention. Up to now, it has been reported that many chemicals such as 2,5-dimethylfuran (DMF) [18,19,20,21], furan-2,5-diyldimethanol (DHMF) [22,23], furan-2,5-dicarbaldehyde [24,25,26], furan-2,5-dicarboxylic acid [27,28,29,30,31], cyclopentanone derivatives [32,33,34,35,36,37,38], acyclic 2,5-diketones [39,40,41] and so on [42,43,44,45,46,47] can be obtained from HMF through hydrogenation, oxidation and other catalytic processes. The acyclic 2,5-diketones, derived from the hydrogenation of HMF, are valuable intermediates with widespread applications in polymers, surfactants, medicine and solvents.

1-hydroxy-2,5-hexanedione (HHD) is a typical 2,5-diketone obtained from HMF [32]. HHD is an intermediate for 3-methyl-2-cyclopenten-1-one (MCP), which is a commercial perfume traditionally obtained from adipic acid via multi-step procedures using organic solvents and toxic reagents with low yield. The use of HMF as a raw material for the production of MCP through HHD as the intermediate is a competitive route, considering the renewable resources, two-step procedure and water as the solvent. As the conversion of HHD to MCP is an intramolecular aldol condensation reaction with high yield at mild conditions, the transformation of HMF to HHD is a key step for this route.

The proposed pathway for the hydrogenation of HMF to HHD is shown in Scheme 1. DHMF is reported to be the intermediate for the preparation of HHD. Tetrahydrofuran-2,5-dimethylethanol (THFDM), 2,5-hexanedione (HD) and DMF are the possible by-products. G. Descotes et al. gave the first report on this transformation with a yield of 60% of HHD using Pt/C and oxalic acid, in 1991 [48]. After that, there were several reports on the hydrogenation of HMF to HHD catalyzed by heterogeneous or homogeneous precious metal catalysts [48,49,50,51,52,53,54,55,56]. Ir or Ru is favored as a homogeneous catalyst and Pd or Au are effective as heterogeneous catalysts. Though possessing the advantage in recycling, heterogeneous catalysts often have lower activity compared with homogeneous catalysts. To make this reaction more efficient and economical, single atomic-site catalysts (Pd_0.02_∧Au/TiO_2_) [49] and none-precious metal catalysts (Ni_2_P) [56] have also been attempted. Acids are reported to be a necessity for this transformation, as the ring-opening of HMF usually needs the assistance of an acid. Mineral acid, hydrochloric acid and phosphoric acid were proved to be an effective acid additive. F. De Campo and F. Jerome et al. reported that the in situ formation of carbonic acid from CO_2_ is a very potent acid site for this reaction. The only insufficient aspect is the pressure that CO_2_ puts forward higher requirements for the reactor [52]. To avoid the application of acid additives and decrease the total pressure, solid acids are selected as the support for the preparation of supported precious metal catalysts. The bi-functional catalysts Pd_0.02_∧Au/TiO_2_ [49], Pd /MIL-101 [57] and Pd/Nb_2_O_5_ [32] were reported to achieve a satisfactory yield of HHD without the addition of acid additives. However, the condensation of intermediates, catalyzed by the acid site, can occur on the surface of bi-functional catalysts to mask the active catalytic site, based on our recent results [57]. This also further prevented the use of fixed-bed reactors for this reaction.

A durable and straightforward catalyst should be developed for the promotion of the transformation of HMF to HHD. In this manuscript, commercial Pd/C was selected as the hydrogenation catalyst. Acetic acid, with lower corrosiveness compared with hydrochloric acid and phosphoric acid, was used as the acid additive. The combination of commercial Pd/C and acetic acid achieved a high yield of HHD from HMF at a high concentration of HMF. The activity of the used Pd/C could be mostly recovered by being washed in hot tetrahydrofuran.

## 2. Results and Discussion

### 2.1. The Role of Acetic Acid

The hydrogenation of HMF to HHD was firstly conducted in water with different amounts of AcOH. When no AcOH was added, the conversion of HMF was 95% with 18% selectivity to HHD (Table 1, Entry 1). This was consistent with the previous reports that the acid was necessary for the high selectivity to HHD. The DHMF was an intermediate for the conversion of HMF to HHD. The 56% selectivity to DHMF showed that the absence of acid reduced the ring-opening of the furan ring. Once a small amount of AcOH was added, the selectivity to DHMF had a drastic decline, while the selectivity to HHD increased from 18% to 66% (Table 1, Entry 2). The weak acidity condition provided by AcOH could catalyze the ring-opening of the furan ring in DHMF. Then, the effect for the concentration of AcOH on the hydrogenation of HMF was studied by the addition of different amounts of AcOH (Table 1, Entry 2–7). It was found that the selectivity to HHD kept between 64% to 71% as the concentration of AcOH increased from 1.4 to 6.6 wt%. The selectivity to DHMF remained very low at all acid additions. The combination of commercial Pd/C and AcOH was a qualified candidate for the conversion of HMF to HHD. The pure AcOH was also tested as a solvent for this transformation. However, only 24% selectivity to HHD was acquired with a 31% conversion of HMF.

### 2.2. The Influence of Temperature and H_2_ Pressure

The effect of temperature and H_2_ pressure was then investigated and the results are presented in Table 2 and Figure 1. The temperature did not have much influence on the conversion but did affect the distribution of the products. When the reaction was performed at 353 K, selectivity to HHD was 25%, while DHMF was 39%. Though 60 mg of AcOH was added, the high selectivity to the intermediate (DHMF) showed that the low temperature inhibited the ring-opening of DHMF. As a result, the selectivity to the hydrogenated product (THFDM) of DHMF increased to 18%. When the temperature rose to 363 K, the selectivity to DHMF decreased to 8%, while THFDM increased to 42%. The selectivity to HHD had a little increase to 30%. As the temperature continued to grow, the selectivity to THFDM decreased as the HHD increased, with low selectivity to DHMF. We could understand the effect of temperature in this way. The temperature had much more of an influence on the ring-opening of DHMF than the hydrogenation of DHMF. The lower temperature both inhibited the isomerization and hydrogenation reaction to have high selectivity to DHMF. When the temperature increased a little, the hydrogenation of DHMF was prior to the conduct to give the increased selectivity to THFDM. When the temperature was raised to a high extent, the ring-opening of DHMF gained priority to give the increased selectivity to HHD. When the temperature rose to 393 K, the selectivity to HHD did not increase, which should be ascribed to the increasing side reactions. In contrary to the effect of temperature, the H_2_ pressure did not have much influence on the selectivity to HHD but influenced the conversion of HMF. The conversion of HMF increased with the increase in H_2_ pressure. The selectivity to HHD kept around 65% when the pressure increased from 1 to 5 MPa.

### 2.3. Yield of HHD from Highly Concentrated HMF

The reported concentration of HMF in the production of HHD was mostly no more than 5 wt%. We wanted to see whether the combination of Pd/C and AcOH could work at a higher concentration. The reaction was conducted using a different amount of HMF and the results are shown in Table 3. When 0.5 mmol (3.1 wt%) of HMF was used as the substrate, the selectivity to HHD was 59%, which was the lowest within the concentration range investigated. The main by-product was HD. This by-product decreased as the concentration of HMF increased. The selectivity to HHD kept around 70% when the amount of HMF was risen from 1 to 2.5 mmol (5.9 to 13.6 wt%). The selectivity to HHD decreased to 65% and 60% when we continued to increase the amount of HMF to 3.0 and 3.5 mmol (15.9 and 18.1 wt%). The experimental results showed that the combination of Pd/C and AcOH could work at a higher concentration. The conversion of HMF decreased with the increase in the concentration of HMF. Based on our previous work, the condensation of intermediates on the active site would occur during the reaction to reduce the activity of Pd/C. The high concentration of HMF promoted this effect to decrease the conversion.

The reaction time was extended to find whether a high yield of HHD could be achieved at a high HMF load (2.5 mmol) and the results are shown in Figure 2. The reaction time was varied from 0.5 to 3.0 h. A 33% conversion of HMF was obtained with 73% selectivity to HHD after being reacted for 0.5 h. The conversion of HMF increased with the increase in reaction time and the selectivity to HHD was maintained around 70%. The highest yield of HHD could reach 66% (93% conversion of HMF and 71% selectivity to HHD) when the reaction time was 3.0 h.

The mole ratio for HMF/Pd of 420 was among the common amount found in the literature for the conversion of HMF to HHD when using supported Pd as catalysts [32,40,49,52,55,57]. To study the influence of the catalyst amount on conversion and selectivity, we performed the experiments using a different amount of catalysts. The results are shown in Figure 3. Lower conversion and selectivity were obtained when the mole ratio of HMF/Pd was 840. As the amount of catalyst increased, the conversion increased rapidly to over 90%, accompanied with the increase in selectivity. The highest selectivity (71%) to HHD was gained at a mole ratio of 420, while the best yield (68%) was obtained at 280. When the mole ratio was lower than 280, the HMF was almost completely converted. The selectivity to HHD decreased slightly as the amount continued to increase. This might be caused by the increase in side reactions. It indicated that an appropriate amount of catalyst was needed for the reaction.

### 2.4. Reusable Performance

The repetitive utilization of the catalyst at a high HMF load (2.5 mmol) was studied by the recovered Pd/C after each reaction (Figure 4). The catalysts were recovered in two ways. The first washed the catalysts by water and ethanol for three repetitions, respectively, at room temperature after filtering (Figure 4a). The other method was further washing the catalysts in hot tetrahydrofuran (393 K) after washing by water and ethanol (Figure 4b). It was found that the activity was greatly reduced, especially for the third use, revealed by the conversion of HMF being decreased from 93% to 40% (Figure 4a) when just washed by water and ethanol (Figure 4a). The Pd/C could not be reused by simply washing. This was probably caused by the condensation of intermediates on Pd at a high HMF load, and thus the polymer could not be cleaned up by water and ethanol at room temperature. In order to be able to reuse the Pd/C, the catalyst was washed in hot tetrahydrofuran (393 K) to dissolve the capped polymer (Figure 4b). After the wash, the tetrahydrofuran changed from colorless to light yellow and most of the activity of Pd/C was recovered. It can be seen from Figure 4b that the conversion of HMF had a very little decrease even for the fourth reuse. The selectivity to HHD kept above 65%.

### 2.5. Characterization

X-ray photoelectron spectroscopy (Figure 5) was used to find the difference between the fresh and used catalysts after the third cycle in Figure 4a. The survey scan showed the signal of C, O and Pd and had no noticeable difference for both catalysts (Figure 5a). After the deconvolution process, the high-resolution spectrum of Pd 3d for fresh Pd/C showed binding energy peaks at 335.5 and 337.3 eV for Pd 3d_5/2_ and 340.9 and 342.6 eV for Pd 3d_3/2_ (Figure 5b). The 335.5 and 340.9 eV results can be attributed to Pd (0), while the other two peaks are the signal of Pd (II) [58,59,60,61]. The Pd (II) should generate in the passivation process of Pd/C. The deconvolution of the used catalyst also gave four peaks with a similar binding energy as the fresh catalyst (Figure 5c). The peak intensity of Pd decreased compared with the fresh catalyst. This was probably caused by the presence of a polymer on the surface of Pd, which was produced by the condensation of the intermediates.

The TEM images and the corresponding distribution of the particle size for the fresh and used Pd/C after the third cycle in Figure 4a are shown in Figure 6. The fresh Pd/C had a narrow particle size distribution of 2–4 nm (Figure 6b). After the reaction, the Pd nanoparticles enlarged a little and some large particles could be observed in the TEM image (Figure 6c). The particle size distribution of used Pd/C increased to 3–5 nm which should be one of the reasons for the decline in activity (Figure 6d). The HRTEM images are shown in the Figure 6e,f. The lattice stripe corresponding to the Pd (111) crystal plane can be clearly seen in the fresh Pd/C (Figure 6e). The diffraction spots corresponding to the Pd (111) crystal plane can also be seen after the fast Fourier transform. The used Pd/C also showed diffraction spots corresponding to the Pd (111) crystal plane after the fast Fourier transform (Figure 6f). However, the lattice stripe was indistinct for the used Pd/C. This was another evidence for the carbon deposition on the Pd surface, which was caused by the condensation of the intermediates.

## 3. Materials and Methods

### 3.1. Materials

Pd/C (5 wt%) was obtained from Shanxi Rock New Materials Co. Ltd. Baoji China. HMF was bought from Zhengzhou Alpha Chemical Co. Ltd. Zhengzhou China, and 2,5-hexanedione, AcOH, ethanol and tetrahydrofuran were purchased from Aladdin Chemistry Co. Ltd. Shanghai China, and 2,5-di(hydroxymethyl)furan was prepared by reducing HMF with sodium borohydride. Unless otherwise stated, the chemicals were used as received.

### 3.2. Characterization

X-ray photoelectron spectroscopy (XPS) was conducted on a Thermo Fisher K-alpha (Waltham, MA, USA) using an Al Ka (1486.6 eV) radiation source. The pass energy for the survey scan and high-resolution spectra were 200.0 and 50 eV, respectively. Transmission electron microscopy (TEM) images were collected by an FEI Tecnai G2 F20 microscope (Hillsboro, OR, USA). The accelerating voltage was 200 kV. Before the test, the sample was ultrasonic in ethanol.

### 3.3. Catalytic Reactions

A 20 mL stainless-steel autoclave equipped with a glass lining was used for the hydrogenation of HMF. A magnetic stirring oil bath with a temperature controller was used to control the temperature. For a typical procedure, the magneton, HMF (1 mmol, 126.1 mg), H_2_O (2.00 g), AcOH (1 mmol, 60.0 mg) and Pd/C were put in the glass lining. Then, the lining was put into the autoclave and the autoclave was purged with 2 MPa H_2_ four times to exclude the air before 4 MPa H_2_ was charged. Then, the autoclave was put in the oil bath at 393 K. After the reaction, the autoclave was cooled down to room temperature. The internal standard and ethanol were added into the glass lining. The mixture was centrifugated and the liquid phase was used for analysis. The reaction was repeated at least twice to calculate the confidence interval with 95% confidence. The solid was washed with water and ethanol for three times respectively and dried in a vacuum for the next run. To recover the activity, the solid was then washed by tetrahydrofuran at 393 K for 1 h after being washed by water and ethanol. The procedure for washing by tetrahydrofuran was performed in a 100 mL stainless-steel autoclave (Gongyi Ying Yu Instrument Factory, Gongyi, China). Before being heated, the air in the autoclave was replaced by 1 atm N_2_ by shlenk technology.

### 3.4. Product Analysis

The qualitative and quantitative analysis was conducted on an Agilent 6890N GC/5973 MS (Agilent Technologies Inc., Santa Clara, CA, USA) and Agilent 7890A GC (Agilent Technologies Inc., Santa Clara, CA, USA), respectively. The detailed process has been described in our previous report [32].

## 4. Conclusions

The combination of commercial Pd/C and AcOH was effective for the conversion of HMF to HHD. The addition of a small amount of AcOH promoted the ring-opening of DHMF, thus leading to the high selectivity to HHD. The reaction could conduct at a high concentration of HMF and a 68% yield of HHD was achieved from a 13.6 wt% aqueous solution of HMF. The activity of the used Pd/C decreased due to the carbon deposition and grain growth of the Pd nanoparticles. The activity of the used Pd/C could be mostly recovered through being washed by hot tetrahydrofuran.

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
