# Peer review of "Preparation of 1-Hydroxy-2,5-hexanedione from HMF by the Combination of Commercial Pd/C and Acetic Acid"

_molecules, 2020, doi:10.3390/molecules25112475_

Round 1

Reviewer 1 Report

The paper of Y. Yang et al on the reduction of HMF to 1-hydroxy-2,5-hexanedione using commercial Pd/C and H2 lack of novelty to be accepted as a paper in Molecules.

The transformation of HMF to 1-hydroxy-2,5-hexanedione has already been studied by G. Descotes et al. (Bull. Soc. Chim. Fr., 1991, 128, 704–711). As one of his numerous students, I'm particularly sensitive that his name should not be forgotten. In this publication cited 69 times (web of science), a yield of 60% of 1-hydroxy-2,5-hexanedione was obtained using Pt/C and oxalic acid.

Later, F. De Campo and F. Jerome published the Pd/C-catalyzed hydrogenation of HMF in water and under CO2 to afford 1-hydroxy-2,5-hexanedione with up to 77% yield (ChemSusChem 2014, 7, 2089 – 2095, DOI: 10.1002/cssc.201402221). It should be noticed that in this publication, CO2 was used as an acidic catalyst (carbonic acid). This publication should not be cited in the middle of several other publications, lines 42-43 “There had several reports on the hydrogenation of HMF to HHD catalyzed by heterogeneous or homogeneous precious metal catalysts [45-53].” The authors should compare their results with this publication.

Furthermore, the authors suggested the replacement of hydrochloric acid by acetic acid because of its corrosiveness (lines 58-59). What about CO2?

As such, the results present in this paper lack of novelty (commercial? Pd/C, acetic acid as catalyst, and classical yield (below 70%) and could not in the present form be accepted in Molecules.

Reviewer 2 Report

This paper deals with the synthesis of 1-hydroxy-2,5-hexanedione (HHD) from a renewable starting reagent, the 5-hydroxymethylfurfural (HMF), using Pd/C as a catalyst and acetic acid as a reagent. With the detailed evaluation of the reaction parameters, the authors clarified why the temperature and the acetic acid concentration are crucial for the selective conversion of HMF into HHD. An efficient recycling procedure for the catalyst was also presented. Then, I recommend this paper for publication in Molecules after the minor points listed below have been properly addressed by the authors.

  • Line 26: The author talked about the acid transformation of carbohydrate to HMF but, as widely demonstrated in the literature (see, inter alia, Chambon, F. et al. (2011) Catal. B-Environ. 105, 171–181; Tallarico, S. et al. (2019) Sci. Rep. 9, 18858), it is possible to perform different degradation pathways, by working under Lewis or Broansted acid conditions, obtaining Lactic acid or HMF respectively. Please, complete this sentence.
  • Lines 35 and 64: please, insert the abbreviation close to the related words.
  • Tables 1-3: please insert the HHD yields together with its selectivity and the HMF conversion values.
  • Section 2.4: Did the author evaluate the Pd leaching? Did the author think to use 2-Methyltetrahydrofurane, instead of THF for the catalyst washing? How is it possible that they use THF at 393 K for 1 h, since its b.p. is 339 K? Please, clarify it, also in the section 3.3.
  • Line 153: The author did not specify how many times was used the Pd/C catalyst before the analysis.
  • Figure 5: Considering the Fig. 5 description (lines 167-171), I think that are present different mistakes both in the figure 5 (seems that Fig. 5b and 5d are completely equal) and in the caption (according to the caption the particle size distribution of fresh Pd/C is the fig. 5c, while the TEM image for the used Pd/C is the Fig. 5b).
  • Ref 20: please check abbreviation.

Reviewer 3 Report

The manuscript entitled “Preparation of 1-hydroxy-2,5-hexanedione from HMF by the combination of commercial Pd/C and acetic acid” by Yang and co-workers describes a simple system for reduction of hydroxymethylfurfural into the product 1-hydroxy-2,5-hexandione. This appears to be a relatively robust catalyst system, and I think the publication can be suitable for Molecules. However, at this time I think there are a number of things that need to be addressed prior to publication:

1) On line 32 the authors state “The 2,5-diketones, derived from the hydrogenation of HMF, were valuable intermediates with widespread applications in polymer, surfactant, medicine and solvent.” It would be more clear to call these “acyclic 2,5-diketones”.

2) On line 37, the authors use the acronym MCP but have not previously defined it.

3) It would be beneficial to move Scheme 1, or at least the contents of it, earlier. Folks that aren’t directly in this research area will likely find the introduction acronym heavy, and it would be better to show the structures when they’re introduced to make it easier to follow.

4) The authors use HAc to represent acetic acid, however this is incorrect chemical terminology. Ac implies an acetyl group, or COCH3. Therefore HAc would be ethanal (acetaldehyde) rather than acetic acid. They should instead change this to HOAc throughout the manuscript.

5) The authors present data in tables 1-3 and figures 1 and 2, but have no error bars. Were all of these the result of a single run, or an average of multiple runs? If the later, error bars should be added, and if not, this really needs to be done. The lack of trends in some areas where all the data is close suggests the data points may all be within experimental error, so having the errors would clarify that point.

6) Figure 3 is a bit misleading. I think either the THF wash needs to be performed for each step, or none, so that a trend can be seen. As is, it is hard to understand why Cycle 2 was less active than 4.

7) Although this is a very interesting paper from the perspective of process chemistry, there are two major limitations to actual implementation. The first of these is that no scale-up is demonstrated. Considering this is a heterogeneous system, does the catalyst still perform at say 10, or 50 times the reaction scale. If not, then the utility is a bit more limited.

8) The second challenge, though probably outside of the scope of this manuscript, is whether or not the procedure can easily be adapted to flow conditions. Though I don’t think the authors need to run any experiments to demonstrate feasibility, I believe a brief review of relevant literature and discussion of the feasibility would be in order.

Round 2

Reviewer 1 Report

Previous works have been correctly cited, but important information on the catalyst is missing. What is the amount of Pd in the catalyst? 0.5 %wt, 1%wt, 10wt%? No information could be found in the text nor the Materials and Methods section. Where did this commercial catalyst was bought? Commercial reference?

The abbreviation of acetic acid should be AcOH rather than HOAc

What is the catalytic amount of Pd used? This information should be also added in the captions of table 1 and figures. Could this value compare to literature? In table 3, the ratio HMF:Pd refer to:
- mass of HMF: mass of Pd
- mass of HMF : mass of Pd/C?

How the optimun catalytic amount of Pd (or mass of catalyst) was determined? An experiment showing the influence of Pd on the conversion and selectivity is clearly needed.

In Table 1, the concentration of AcOH should also be added rather than the mass of added AcOH, the same is true from HMF in Table 3. Concentration is a more important requirement than the mass of reactant in the reactor.

Reviewer 3 Report

I appreciate the work that the authors put into revising this manuscript. It is much clearer now the utility. I cannot speak to Reviewer 1's points about novelty, but I think this is now appropriate for publication.

Round 3

Reviewer 1 Report

Essential modifications regarding the amount of Pd have been added to the manuscript.

It is claimed in the paper line 153 that "The mole ratio for HMF:Pd of 420 was among the common amount in literature for the conversion of HMF to HHD when using supported Pd as catalysts." but no reference was given. Some references should be added.

line 253: The activity of used Pd/C "decreased due to" instead of "reduced a lot caused by"

line 254: The activity of used Pd/C was not recovered. The conversion decreased from 92 to 78% in only 4 cycles and the authors do not show a stabilization of the conversion.
